# Sleep Duration and Effort-Reward Imbalance (ERI) Associated with Obesity and Type II Diabetes Mellitus (T2DM) among Taiwanese Middle-Aged Public Servants

**DOI:** 10.3390/ijerph17186577

**Published:** 2020-09-09

**Authors:** Dann-Pyng Shih, Ping-Yi Lin, Wen-Miin Liang, Po-chang Tseng, Hsien-Wen Kuo, Jong-Yi Wang

**Affiliations:** 1Department of Public Health, China Medical University, Taichung 40402, Taiwan; 90930@cch.org.tw; 2Center for Teaching Excellence, Changhua Christian Hospital, Changhua 50006, Taiwan; 3Transplant Medicine & Surgery Research Centre, Changhua Christian Hospital, Changhua 50006, Taiwan; 69221@cch.org.tw; 4Department of Medical Research, China Medical University Hospital, Taichung 40447, Taiwan; 5Department of Health Services Administration, China Medical University, Taichung 40402, Taiwan; wmliang@mail.cmu.edu.tw; 6Health Promotion Administration, Ministry of Health and Welfare, Taipei 10341, Taiwan; pochang@hpa.gov.tw; 7Institute of Environmental and Occupational Health Sciences, National Yang Ming University, Taipei 11221, Taiwan; 8School of Public Health, National Defense Medical Center, Taipei 11490, Taiwan

**Keywords:** sleep duration, job strain, obesity, type 2 diabetes mellitus (T2DM)

## Abstract

(1) Limited evidence has shown the mediating effects of work characteristics and sleep duration on obesity and type 2 diabetes mellitus (T2DM) among adults. The objective of this study is to assess the interaction effects between sleep duration and effort–reward imbalance (ERI) on the risk of obesity and T2DM among Taiwanese public servants aged 40–60. (2) A national survey for Taiwanese public servants was conducted by multistage stratified random cluster sampling based on proportional probabilistic sampling. A total of 11,875 participants aged 40–60 years old were collected; (3) 3.6% of participants had self-reporting T2DM diagnosed by a physician and the prevalence of overweight and obesity were 44.0% and 15.8%, respectively. There was a significant correlation between sleep hours for the workday and risk of T2DM in non-obese and obese groups (odds ratio, OR = 1.48 and 1.39, respectively), but this did not exist for the weekend/vacation group. Similar trends in the two groups by sleep hours on a workday, obesity and overweight were significantly associated with the risks of T2DM. Clearly, sleep duration and ERI were moderating factors on the association between BMI and on the prevalence of T2DM. (4) A short sleep duration and heavy job stress contributes to the risk of weight gain and T2DM development.

## 1. Introduction

Epidemiological evidences have shown an increasing prevalence of type 2 diabetes mellitus (T2DM) globally. The proportion of people with T2DM increased by 72%, from 194 million people to 333 million, between 2003 to 2025 based on the prediction of the International Diabetes Federation [1]. In Taiwan, overall 5-year prevalence of T2DM from 1992–1996 for men and women were 187.1 and 218.4 per 100,000 population, respectively, and prevalence of obesity increased from 39.2% in 1992 to 47.6% in 1996 [2]. The population is aging rapidly in Taiwan along with an increasing percentage of a sedentary lifestyle and a high-calorie dietary intake. The prevalence of diagnosed T2DM increased significantly from 4.9% in 1985 to 5.79% in 2000 and to 8.30% in 2007. Over an eight-year period in Taiwan, the prevalence of T2DM increased by 43.35%, which was higher than the global prediction of a 39% increase from 2000 to 2030 based on the World Health Organization (WHO) [3,4]. Evidence has demonstrated that frequent consumption of sugar-sweetened beverages has contributed to adverse health consequences including obesity, metabolic syndrome (MetS), T2DM, and non-alcoholic fatty liver disease [5].

Furthermore, previous studies [6,7,8,9] have indicated that decreased sleep duration and quality were associated with an increase in weight gain and adiposity, revealing that they may be contributing factors toward obesity and other chronic diseases, including MetS, cardiovascular diseases (CVDs), and T2DM. However, the results have been inconsistent [10]. In addition, long-term exposure to stressful work may be a contributor to an increased risk of obesity and other metabolic diseases, which can result from alterations in the hypothalamic–pituitary–adrenal (HPA) axis [11]. Public servants frequently encounter higher perceived job stress, poor sleep quality, and more persistent fatigue problems due to external competition and the downsizing of governmental workforces. In addition, they have irregular and imbalanced diets, and sedentary work schedules, which may contribute to the occurrence of obesity and T2DM. Studies have not yet shown the effects of psychosocial stress in the workplace and sleep duration on obesity and T2DM among adults, especially public servants. Work-related stress and T2DM has been assessed, but the association between job stress and T2DM remains unclear [12]. The objective of this study is to assess the interaction effects between sleep duration and effort–reward imbalance (ERI) on obesity and T2DM among Taiwanese public servants aged 40 to 60. In addition, the study will assess the association between obesity and T2DM by moderating the effect of ERI or sleep duration.

## 2. Materials and Methods

### 2.1. Study Population

Taiwan’s Health Promotion Administration (HPA) launched a national survey for public servants using multistage stratified random cluster sampling, based on proportional probabilistic sampling (PPS). A total of 10,795 participants from 647 registered governmental institutions were enrolled and the overall response rate was 32.5%. However, most people were not willing to participate due to limited time available and vacations. Eligible public servants, defined as formally registered and employed by the central and local government, anonymously and voluntarily filled out the web-based questionnaire and gave informed consent at the time of the study. Participants with ages lower than 40 and higher than 60 were excluded due to short work duration and seniority. Reasons of non-response included vacations, requested time off, or insufficient time to fill out the questionnaire, and these do not affect the objective of this study. Ethics approval and consent to participate: the study had ethical approval from the China Medical University (CMUH105-REC3-091).

### 2.2. Measurements

The questionnaire included demographic data, anthropometric measurements (height, weight, and wrist circumference), work schedule, lifestyle habits (tobacco smoking, alcohol consumption, betel nut chewing, and frequency of physical activity), sleep duration during the workday and on weekends/vacations, and history of diseases. Sleep duration was self-reported for the workday and weekend/vacation and was classified into two groups, with short sleep duration (SSD) as ≤6 h and long sleep duration (LSD) as ≥7 h [6]. Two types of sleep duration were combined for workday and weekend/vacation. Levels of sleep duration were classified into four levels as follows: level 1 represented LSD for workday and LSD for weekend/vacation; level 2 represented LSD for workday and SSD for weekend/vacation; level 3 represented SSD for workday and LSD for weekend/vacation; level 4 represented SSD for workday and SSD for weekend/vacation. A Chinese version of the Job Content Questionnaire (C–JCL) was developed and validated for reliability and validity for blue-collar workers [13]. Effort–reward imbalance (ERI) was measured with a short-version modified questionnaire [14], which contained 3 items for measuring effort and 7 items for measuring reward. The ERI ratio for each study participant was computed using the sum scores for efforts as the numerator (Effort) and the sum scores for rewards as the denominator (Reward) multiplied by a correction factor of 0.43 [(Effort/3)/(Reward/7)]. An ERI ratio >1 indicated an exposure to high ERI at work, which also constituted as a perceived psychosocial work stress. Cronbach’s alpha coefficients for each scale in the ERI model ranged from 0.65–0.88, which is consistent with Siegrist [14] of 0.61–0.91. Some public servants (n = 440) were given two questionnaires using both an in-person interview and a computer-based questionnaire. The highly consistent results were obtained from both methods of questionnaire, indicating that our findings from the web-questionnaire were reliable. Because in the beginning we doubted that there might be a discrepancy in different groups with high recovery or low recovery, a small portion of participants were selected to validate the consistent results from the two groups of low and high recovery. Our findings did not show a significant difference between the two groups.

Overweight and obesity were defined by using body mass index (BMI), which was calculated by dividing weight by height (kg/m^2^). Based on Taiwan’s guideline for adult men and women from Chiu’s study [15], a BMI of 18.5–24 is considered healthy, a BMI of 24–27 is overweight, a BMI of 27–30 is mildly obese, a BMI of 30–35 is moderately obese, and a BMI exceeding 35 is severely obese. Participants who reported having T2DM were diagnosed by a physician or had at least one hospital admission with a diagnostic code of T2DM. Each participant was required to be treated or regularly monitored after the government employment.

### 2.3. Statistical Analysis

SPSS 24 package (IBM Corp, Armonk, NY, United States) was used to analyze the data. Univariate analysis was used to examine the difference between two groups of sleep hours with demographic data and work characteristics. Chi-square test was used to examine the sleep duration for workday and weekend/vacation and correlations with obesity, overweight, BMI, and ERI. Using multiple logistic regression adjusted for age, gender, education level, marital status, smoking and alcohol consumption, levels of sleep duration on workdays and on weekends/vacations were respectively examined for correlations with the percentages of obesity/overweight and T2DM occurrence. Odd ratios (ORs) of T2DM that were correlated with obesity, overweight, and BMI were stratified into two groups of ERI using multiple logistic regression adjusted for covariates. The interaction effect of ERI and sleep duration on overweight, obesity, and T2DM was measured by relative excess risk using synergy index (S) (available at website with http://www.epinet.se.).

## 3. Results

Table 1 shows demographic information and two groups of sleep durations for the workday. There was only a significant difference on sleep duration in terms of gender; 56.2% of women were in the SSD group, which was higher than the 43.8% for men. Other demographic information had no significant difference for sleep duration. In Table 2, two groups of sleep duration for workdays and weekends/vacations were significantly associated with obesity, overweight, BMI, and ERI. There was a higher percentage of people with obesity and overweight for SSD for workday and weekends/vacations. Levels of BMI were significantly associated with sleep duration for workdays and weekends/vacations. ERI levels were significantly associated with sleep duration for workdays and weekends/vacations. A total of 55.1% and 52.6% had ERI >1 for SSD for workday and weekends/vacations, respectively.

Table 3 indicates the associations between sleep duration during the workday and on weekends/vacations with T2DM using multiple logistic regression, adjusted for age, gender, education level, marital status, and smoking and alcohol consumption. People with SSD for workdays and weekends/vacations had respectively 1.52 and 1.23 times the ORs of T2DM, but there was only a statistical significance for workdays. Four kinds of sleep duration were significantly associated with T2DM. People with SSD for both workday and weekend/vacation (level 4) had 1.48 times OR of T2DM. In addition, the OR of T2DM was 1.54 times greater for the group with SSD in the workday and LSD on weekends/vacations (level 3).

Four types of sleep duration were defined as follows: level 1 represented LSD for workday and LSD for weekend/vacation; level 2 represented LSD for workday and SSD for weekends/vacations; level 3 represented SSD for workday and LSD for weekend/vacation; level 4 represented SSD for workdays and SSD for weekends/vacations.

Table 4 shows associations between sleep duration for workdays and weekends/vacations and T2DM in non-obese and obese groups using multiple logistic regression after adjusting for age, gender, education level, marital status, and smoking and alcohol consumption. There was a significant association between sleep duration for workday and T2DM (OR = 1.48 for the obese group and OR = 1.39 for the non-obese group, respectively), but not for weekends/vacations. For the group with SSD in workdays and a LSD on weekends/vacations (level 3), the prevalence of T2DM were 1.52 times greater for the obese group and 1.44 times greater for the non-obese group. Similarly, there was 1.43 and 1.33 times the ORs of T2DM for obese and non-obese groups, respectively, for the people with SSD for both workdays and weekends/vacations (level 4).

Table 5 indicates that ORs of T2DM were significantly associated with obesity, overweight, and BMI for SSD and LSD groups for workdays using multiple logistic regression adjusted for gender, age, smoking, alcohol consumption, and betel nut chewing. There were similar trends on ORs of T2DM with obesity and overweight between the SSD and LSD groups for workdays. BMI were significantly associated with T2DM development in both groups of SSD and LSD. There were considerably higher ORs (OR = 7.27 and OR = 9.48) of T2DM for those with BMI levels of 30–35 and >35 in the LSD during workday group. Clearly, there are interaction effects of BMI and sleep duration on T2DM.

Table 6 shows that T2DM is associated with obesity, overweight, and BMI stratified by the two ERI groups for workdays using multiple logistic regression adjusted for gender, age, smoking, alcohol consumption, and betel nut chewing. For all participants, there were 3.76 and 3.10 times the ORs of T2DM for those with obesity and overweight, and dose-dependently associated with levels of BMI. Participants with a BMI higher than 35 had the highest OR (7.02 times) of T2DM. The ORs of T2DM were significantly associated with obesity, 4.34 times greater for the group with ERI ≤ 1 and 3.19 times greater for the group with ERI > 1. Similarly, there was a 2.86 and 3.33 times greater OR of T2DM for those who were overweight for both ERI groups. There were consistently higher OR of T2DM with levels of BMI in the group with ERI > 1 compared to the group with ERI ≤ 1. The BMI ≥ 35 group had a 34.37 times greater OR of T2DM in the ERI>1 group, which was higher than the ERI ≤ 1 group (OR = 2.10). There were considerably different effects between BMI and the ORs of T2DM in the two groups of ERI, indicating that ERI may act as a moderating factor to obesity and T2DM.

Table 7 indicated the interaction effect of ERI and sleep duration during the workday on overweight/obesity and T2DM using multiple logistic regression after adjusting for age, gender, education level, marital status, and smoking and alcohol consumption. Interaction effects of ERI and sleep duration during the workday on overweight and obesity were found, with a synergy index of 2.36 and 1.30, respectively. The group with ERI > 1 and SSD had the highest ORs of overweight (OR = 1.33, *p* < 0.01) and obesity (OR = 1.39, *p* < 0.01). There were significant differences between ERI and sleep duration on T2DM prevalence in the non-obese and obese groups. For the obese group, the ERI > 1 and SSD group had the highest OR of T2DM (OR = 1.97), followed by the ERI ≤ 1 and SSD group (OR = 1.78) and the ERI > 1 and LSD group (OR = 1.70). For the non-obese group, there was only a significantly higher OR of T2DM for the ERI ≤ 1 and SSD group (OR = 1.79).

## 4. Discussion

The study indicated that public servants with SSD (≤6 h) during the workday and on weekends/vacations had a higher prevalence of overweight/obesity. This is consistent with the previous study on police officers in Taiwan [16], which reported that those who slept less than 5 h were more likely to experience abdominal obesity in Taiwan, and those with higher scores of sleep disturbances had a higher prevalence of metabolic syndrome (MetS). In addition, short sleep duration has been reported to be associated with increased insulin resistance, but Chang, et al. [17] also reported that longer sleep duration was an independent risk factor associated with increased insulin resistance in vegetarians. Obesity, work characteristics, and sedentary lifestyle were widely known to be associated with the risk of T2DM, which represents up to 90% of prevalent cases of diabetes [18,19]. In this study, intensity of job stress was measured by ERI, which indicted the state of physical and mental strain caused by an imbalance between strict job requirements and an inadequate ability to adapt and cope. We found that ERI acted as a contributing factor for obesity and T2DM. BMI was significantly associated with prevalence of T2DM for both ERI and sleep duration. Importantly, the greater prevalence of T2DM (OR = 34.37) was found in the group with a BMI higher than 35 as well as in the group with a high level of ERI. Research showed that psychosocial risk in the workplace was independently associated with mental health problems and job satisfaction was a contributor to stress-related mental health problems among Japanese public servants [20]. However, a systematic review and meta-analysis of cohort studies showed that work-related stress had no significant association with the risk of T2DM, with a relative risk (RR) of 0.94 for job demands, 1.16 for decision latitude, and 1.12 for job stress. Since the findings from previous studies on work-related stress on the risk of T2DM were inconsistent and inconclusive, it can be assumed that the etiology of T2DM is multifactorial and work-related stress may contribute to, or increase, the risk of T2DM development [21]. Similarly, a study in the US on working adults 50 years old and older found that high work strain and passive occupations were associated with an increased risk of T2DM [22]. Job stress was associated with a lower risk of T2DM in non-obese men but not in obese individuals, and an inverse relationship with a higher risk of T2DM was found in obese women but not in non-obese women [23]. It can be seen that gender and obesity play a critical role in determining the association between work stress and T2DM. Although it is well-known that job stress is a risk factor for obesity-related disorders such as cardiovascular disease and T2DM, more research is required to better understand how psychosocial work factors link with the risk of T2DM. Besides, the other risk factors for T2DM, including obesity, lifestyle habits, and work characteristics, need to be considered in the study design and data analysis.

Our findings assessed the relationship between sleep duration and job stress on the prevalence of T2DM after adjusting for obesity. There was a considerably higher OR (OR = 7.27 and OR = 9.48) of T2DM for the group with BMI levels of 30–35 and >35 and for sleep duration ≥7 h during the workday. Clearly, there was an interaction effect of BMI and sleep duration on the prevalence of T2DM. In the study, SSD and ERI were the main contributors towards the prevalence of T2DM in the non-obese and obese groups. ERI and SSD acted as a modifier for obesity and the prevalence of T2DM. Although accumulating evidence has indicated that sleep loss may increase the prevalence of obesity and T2DM, there have been few studies that have assessed the moderating effects of both sleep loss and job stress on T2DM in non-obese and obese groups. In addition, the study found that levels of sleep duration (1–4) were dependently associated with T2DM, which explained that insufficient sleep duration both on workdays and on vacation included higher risks of metabolic disease. The findings are consistent with a Nurses Health study, which showed an increased risk of symptomatic T2DM among those reporting sleep durations of 5 h or less, compared with 7–8 h, after controlling for BMI, shiftwork, hypertension, exercise, and depression [24]. This is similar to findings from large US samples that revealed that those sleeping five hours or less, or sleeping nine or more, were at an increased risk of developing T2DM [25]. Short (<6 h) and long (≥8.5) sleep duration on ORs of T2DM were 1.55 (95%CI = 1.07–2.24) and 2.83 times greater (95%CI = 1.19–6.73), respectively. A U-shaped relationship between sleep duration and metabolic syndrome and diabetes has been observed [26]. The mechanisms underlying this association may be a discrepancy between short and long sleep duration because there is a less clear indication of possible mechanisms mediating the effect of a long duration of sleep as a cause of T2DM [27].

There is sufficient evidence for a causal link between sleep duration and job stress on the adverse effects of metabolic functions, although its mechanisms are not well understood. It has been hypothesized that sleep duration or job stress could alter appetite regulation, increase body weight, decrease glucose tolerance, increase blood pressure, and increase insulin resistance and sympathetic tone [28,29]. Moreover, it is postulated that biological mechanisms occur through an upregulation of the neuroendocrine control of appetite to reduce leptin and increase ghrelin levels [30]. Changes in endocrinal functioning that increase cortisol concentrations and sympathetic nervous system activity could result in weight gain and a higher risk of obesity [31]. It is possible that persons exposed to SSD and a heavy workload become more fatigued, which may also result in lower levels of energy expenditure, particularly in sedentary work that promotes physical inactivity for public servants. Because SSD has detrimental effects on health T2DM, these insights may help in the development of new preventative and therapeutic approaches combating obesity and T2DM and help improve the quality and/or quantity of sleep and to modify the workloads of public servants [32]. A SHISO cross-sectional study on Chinese workers showed that levels of over-commitment and ERI may increase the risk of dyslipidemia [33], but there was no significant correlation between job stress and glycosylated hemoglobin (HbA1c) for men [34].

Our study has several limitations in its ability to assess the causal relationship between sleep duration and ERI on T2DM in obese and non-obese groups. First, all data from public servants were collected by a self-reported web questionnaire. This may cause a misrepresentation of T2DM diagnoses and causes of sleep duration. Nevertheless, self-reported history of T2DM diagnosis was previously shown to be reliable [35]. Using objective measurements of sleep were not practical and often not feasible in a nationwide survey. A previous study [36] showed good associations between subjective estimates of sleep duration and direct assessments by actigraphy and polysomnography. Secondly, healthy worker effects may occur and lead to an underestimation of the effects due to public servants voluntarily participating in the national survey. Response rate in the study was 32.5% due to vacations, requested time off, or limited time to fill out the questionnaire, and these do not affect the objective of the study. There were no significant difference in demographics and work characteristics in the responders and non-responders. Since some public servants were laid off or retired early due to health problems or were poorly adapted to government employment, the study’s participants were likely to be in good health, with a lower incidence of T2DM. However, our findings may not apply to other occupations with a heavy workload and repetitive jobs. It is possible to increase the risk of adverse effects. Thirdly, residual confounding and bias remain a possibility due to other contributing factors in the study that were not measured accurately, such as diet intake, sleep quality, and genetic factors. Not all participants were diagnosed with T2DM by measuring their A1C, but they had access to good medical resources to receive medical checkups in Taiwan. Moreover, some evidence indicated that obesity and diabetes risk in adulthood was related to in utero exposure to environmental chemicals, particularly endocrine-disrupting chemicals [37]. Our results assumed that non-measured contributing factors may not be associated with sleep duration and ERI, which may lead to an underestimation due to non-differential misclassification, such as obese people with sleep apnea syndrome. Further research is needed to elaborate on the causal effect of ERI or sleep duration and T2DM by mediating the effect of obesity. The direct and indirect effects of ERI or sleep duration on obesity and T2DM should be assessed using a structure equation model.

This study’s strengths included the first use of a nation-wide survey to examine the effects of sleep duration and job stress on the risk of T2DM in obese and non-obese public servants in Taiwan. Our findings indicated that SSD and ERI might play a contributing role in the relationship between obesity and T2DM. Since public servants faced external competition and workplace downsizing, they experienced higher perceived stress, poor sleep quality, and more persistent fatigue problems. These effects may result in a higher risk of chronic non-communicable disease occurrence and development. An effective health promotion program should be initiated to prevent T2DM development in various jobs, not only for public servants.

## 5. Conclusions

Short sleep duration and heavy job stress contribute to the risk of weight gain and T2DM development. Further research is needed to elucidate the role of sleep quality and job stress on obesity and metabolic diseases based on prospective studies, as well as to implement intervention programs to reduce workload, increase sleep duration, and improve sleep quality for public servants.

## Figures and Tables

**Table 1 ijerph-17-06577-t001:** Demographic information and two groups of sleep durations for workdays.

Items	Sleep Hours ≤ 6 h (N = 4821) *n* (%)	Sleep Hours ≥ 7 h (N = 5974) *n* (%)	*p*
Gender			<0.0001
Men	2073 (43.0)	2795 (46.8)	
Women	2748 (57.0)	3179 (53.2)	
Age (years)			0.3653
40–49	2816 (58.4)	3541 (59.3)	
50–60	2005 (41.6)	2433 (40.7)	
Education level			0.0064
Senior high school	277 (5.8)	408 (6.8)	
College	1144 (23.7)	1278 (21.4)	
Undergraduate	2132 (44.2)	2664 (44.6)	
Graduate	1268 (26.3)	1624 (27.2)	
Marital status			0.6221
Unmarried	748 (15.5)	928 (15.5)	
Married	3995 (82.9)	4963 (83.1)	
Other	78 (1.6)	83 (1.4)	
Smoking			0.617
No	3980 (82.6)	4956 (83)	
Yes	841 (17.4)	1018 (17)	
Alcohol consumption			0.9802
No	4173 (86.6)	5172 (86.6)	
Yes	648 (13.4)	802 (13.4)	
Betel nut chewing			0.1307
No	4537 (94.1)	5662 (94.8)	
Yes	284 (5.9)	312 (5.2)	

**Table 2 ijerph-17-06577-t002:** Two groups of sleep duration for workdays and weekends/vacations and association with obesity, overweight, BMI, and effort–reward imbalance (ERI).

Items	Workday		Weekend/Vacation	
	Sleep Hours <6 h (N = 5244) *n* (%)	Sleep Hours >7 h (N = 6603) *n* (%)	Sleep Hours <6 h (N = 1807) *n* (%)	Sleep Hours >7 h (N = 9989) *n* (%)
Obesity				
No	4312 (82.2%)	5611 (85.6%)	1483 (82.1%)	8440 (84.5%)
Yes	932 (17.8%)	941 (14.4%)	324 (17.9%)	1549 (15.5%)
Overweight				
No	2848 (54.3%)	3723 (56.8%)	950 (52.6%)	5621 (56.3%)
Yes	2396 (45.7%)	2829 (43.2%)	857 (47.4%)	4368 (43.7%)
BMI (kg/m^2^)				
≤18.4	207 (3.9%)	209 (3.2%)	68 (3.8%)	348 (3.5%)
18.5–24	2641 (50.4%)	3514 (53.6%)	882 (48.8%)	5273 (52.8%)
24–27	1464 (27.9%)	1888 (28.8%)	533 (29.5%)	2819 (28.2%)
27–30	612 (11.7%)	680 (10.4%)	215 (11.9%)	1077 (10.8%)
30–35	244 (4.7%)	200 (3.1%)	79 (4.4%)	365 (3.7%)
≥35	76 (1.4%)	61 (0.9%)	30 (1.7%)	107 (1.1%)
ERI				
≤1	2355 (44.9%)	3815 (58.3%)	854 (47.4%)	5316 (53.2%)
>1	2887 (55.1%)	2728 (41.7%)	947 (52.6%)	4668 (46.8%)

**Table 3 ijerph-17-06577-t003:** Sleep hours on workdays and on weekends/vacations associated with T2DM using multiple logistic regression adjusted for age, gender, education level, marital status, smoking and alcohol consumption.

Items	T2DM *n* (%)	Crude OR (95% CI)	Adjusted OR (95% CI)
Sleep hours in workday			
≥7	192 (2.9)	1	1
≤6	232 (4.2)	1.48 ** (1.21–1.80)	1.52 ** (1.25–1.86)
Sleep hours in weekend/vacation			
≥7	332 (3.3)	1	1
≤6	83 (4.6)	1.40 ** (1.10–1.80)	1.23 (0.96–1.58)
Types of sleep duration			
1	186 (2.9)	1	1
2	6 (4.9)	1.75 (0.76–4.23)	1.19 (0.43–3.27)
3	146 (4.1)	1.44 ** (1.15–1.80)	1.54 ** (1.24–1.94)
4	77 (4.5)	1.61 ** (1.23–2.11)	1.48 ** (1.12–1.94)

Note. OR—odds ratio, T2DM—type 2 diabetes mellitus, ** *p* < 0.01.

**Table 4 ijerph-17-06577-t004:** Sleep hours on workdays and on weekends/vacations associated with T2DM using multiple logistic regression based on obesity adjusted for age, gender, education level, marital status, and smoking and alcohol consumption.

Items	Obesity	Non-Obesity
	OR (95% CI)	OR (95% CI)
Sleep hours on workdays		
≥7	1	1
≤6	1.48 ** (1.15–1.92)	1.39 * (1.01–1.92)
Sleep hours on weekends/vacations		
≥7	1	1
≤6	1.20 (0.86–1.67)	1.15 (0.77–1.72)
Types of sleep duration		
1	1	1
2	1.03 (0.25–4.23)	1.36 (0.30–6.23)
3	1.52 ** (1.14–2.02)	1.44 ** (1.00–2.07)
4	1.43 * (1.00–2.04)	1.33 (0.86–2.07)

Note. ** *p* < 0.01, * *p* < 0.05, OR—odds ratio, CI—confidence interval.

**Table 5 ijerph-17-06577-t005:** T2DM associated with obesity, overweight, and BMI based on sleep duration during the workday using multiple logistic regression adjusted for gender, age, smoking, alcohol consumption, and betel nut chewing.

Items	Sleep Hours ≤6 h (N = 5272)	Sleep Hours ≥7 h (N = 6603)	Total
	OR (95% CI)	OR (95% CI)	OR (95% CI)
Obesity (No = ref.)	3.74 **	3.62 **	3.76 **
	(2.81–4.97)	(2.66–4.93)	(3.05–4.63)
Overweight (No = ref.)	3.16 **	2.94 **	3.10 **
	(2.31–4.32)	(2.11–4.10)	(2.47–3.90)
BMI (≤18.4 kg/m^2^ = ref.)			
18.5–24	1.04	1.49	1.17
	(0.37–2.90)	(0.36–6.18)	(0.51–2.68)
24–27	2.10	2.87	2.33 *
	(0.75–5.89)	(0.69–11.97)	(1.01–5.37)
27–30	5.68 **	7.16 **	6.12 **
	(2.12–15.98)	(1.62–30.01)	(2.65–14.14)
30–35	4.64 **	7.27 **	5.59 **
	(1.54–13.97)	(1.62–32.69)	(2.30–13.54)
>35	5.49 **	9.48 **	7.02 **
	(1.59–19.01)	(1.84–48.77)	(2.63–18.81)

Note. ** *p* < 0.01, * *p* < 0.05, BMI—body mass index.

**Table 6 ijerph-17-06577-t006:** Obesity/overweight associated with T2DM in two groups of ERI using multiple logistic regression adjusted for gender, age, smoking, alcohol consumption, and betel nut chewing.

Items	ERI ≤ 1 (N = 6170)	ERI > 1 (N = 5615)	Total
	OR (95%CI)	OR (95%CI)	OR (95%CI)
Obesity (No = ref.)	4.34 **	3.19 **	3.76 **
	(3.24–5.82)	(2.36–4.30)	(3.05–4.63)
Overweight (No = ref.)	2.86 **	3.33 **	3.10 **
	(2.08–3.92)	(2.39–4.63)	(2.47–3.90)
BMI (≤18.4 kg/m^2^ = ref.)		
18.5–24	0.63	3.89	1.17
	(0.25–1.60)	(0.53–28.31)	(0.51–2.68)
24–27	1.04	9.04 *	2.33 *
	(0.41–2.68)	(1.24–65.81)	(1.01–5.37)
27–30	3.45 **	19.05 **	6.12 **
	(1.35–8.85)	(2.60–139.6)	(2.65–14.14)
30–35	4.04 **	12.90 *	5.59 **
	(1.47–11.11)	(1.67–99.9)	(2.30–13.54)
≥35	2.10	34.37 **	7.02 **
	(0.54–8.15)	(4.28–427.2)	(2.63–18.81)

Note. ** *p* < 0.01, * *p* < 0.05, BMI—body mass index, ERI—effort–reward imbalance.

**Table 7 ijerph-17-06577-t007:** Interaction effect of ERI and sleep duration during the workday on overweight/obesity and T2DM using multiple logistic regression after adjusting for age, gender, education level, marital status, and smoking and alcohol consumption.

ERI	Sleep Duration	Overweight	Obesity	T2DM in Obese Group	T2DM in Non-Obese Group
		OR (95%CI)	OR (95%CI)	OR (95%CI)	OR (95%CI)
≤1	≥7	1	1	1	1
>1	≥7	1.06	1.02	1.70 **	1.33
		(0.95–1.18)	(0.88–1.17)	(1.17–2.48)	(0.82–2.14)
≤1	≤6	1.08	1.28 **	1.78 **	1.79 **
		(0.95–1.20)	(1.11–1.48)	(1.22–2.58)	(1.16–2.78)
>1	≤6	1.33 **	1.39 **	1.97 **	1.37
		(1.20–1.48)	(1.22–1.59)	(1.37–2.83)	(0.88–2.15)
Synergy index		2.36	1.30	0.66	0.33

Note. ** *p* < 0.01.

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
