# Peer review of "Sleep Duration and Effort-Reward Imbalance (ERI) Associated with Obesity and Type II Diabetes Mellitus (T2DM) among Taiwanese Middle-Aged Public Servants"

_ijerph, 2020, doi:10.3390/ijerph17186577_

Round 1
Reviewer 1 Report
The objective of this study is to assess the interaction effects of sleep duration and effort-reward imbalance (ERI) on the risk of obesity and T2DM among Taiwanese public servants. There was a significant correlation between sleep hours for the workday and risk of T2DM in non-obese and obese groups, respectively, but it did not exist for the weekend/vacation group. Similar trends in two groups by sleep hours in workday, obesity and overweight were significantly correlated with the risks of T2DM. Clearly, sleep duration and ERI were moderating factors on the correlation between BMI and on the risk of T2DM. Short sleep duration and heavy job strain contribute to the risk of weight gain and T2DM development.
The authors conclude Short sleep duration and heavy job strain contribute to the risk of weight gain and T2DM development. The findings are interest, however, several concerns remain.
- a BMI of 24-27 is overweight, a BMI of 27-30 is mildly obese, a BMI of 30-35 is moderately obese, and a BMI exceeding 35 is severely obese. What is a rationale of the cut-off point?
- Sleep duration was self-reported for the workday and weekend/vacation and was classified into two groups, with short sleep duration (SSD) as ≦6 hours and long sleep duration (LSD) as ≧7 hours. What is a rationale of the cut-off point?
- In obese groups, the patients with sleep apnea syndrome might be involved.Did SAS influence the present result?
- In Table 3,4, sleep hours in workday and weekend/vacation correlated with T2DM. Did levels of sleep duration 1-4 level-dependently correlate with T2DM? Authors should discuss the point.
- Short sleep duration and heavy job strain contribute to the risk of weight gain and T2DM development. It would be better to discuss the mechanism of SSD and the risk of weight gain in greater detail.
Author Response
Point 1: a BMI of 24-27 is overweight, a BMI of 27-30 is mildly obese, a BMI of 30-35 is moderately obese, and a BMI exceeding 35 is severely obese. What is a rationale of the cut-off point?

Response 1: The cut-off point of BMI was based on the Health Promotion Administration (HPA) in Taiwan in Chiu’s study [15]. The definition of obesity and overweight in Asian countries was different from that in Western countries. Please see Line 105-106 in the revised manuscript.
Point 2: Sleep duration was self-reported for the workday and weekend/vacation and was classified into two groups, with short sleep duration (SSD) as ≦6 hours and long sleep duration (LSD) as ≧7 hours. What is a rationale of the cut-off point?
Response 2: Thank you for your comment. The cut-off point of sleep duration was taken from Patel et al., [6] which classified sleep duration into four categories: < 6 hrs, 6 to < 7 hrs, 7 to < 8 hrs, and ≥ 8 hrs. Therefore, in the study sleep duration was just classified into two defined groups of SSD (≦6 hours) and LSD (≧7 hours). Please see Line 84-85 in the revised manuscript.
Point 3: In obese groups, the patients with sleep apnea syndrome might be involved. Did SAS influence the present result?
Response 3:: Thank you for your suggestion. However, civil servants with sleep apnea syndrome were not assessed to be excluded in the study. Therefore, we added the illustration in the limitation. Please see Line 292-293 in revised manuscript. A Chen study (2016) in Taiwan reported that the prevalence of witnessed apnea during sleep was 2.6 %, 3.4 % in males and 1.9 % in females. Our finding assumed that the objective was not affected by the prevalence of sleep apnea syndrome in the civil servants.
Point 4: In Table 3, 4, sleep hours in workday and weekend/vacation correlated with T2DM. Did levels of sleep duration 1-4 level-dependently correlate with T2DM? Authors should discuss the point.
Response 4: Thank you for your suggestion. We have discussed the association between level of sleep duration and prevalence of T2DM. Please see Line 248-250 in the revised manuscript.
Point 5: Short sleep duration and heavy job strain contribute to the risk of weight gain and T2DM development. It would be better to discuss the mechanism of SSD and the risk of weight gain in greater detail.
Response 5: Response: Thank you for your suggestion. The possibility of mechanism was added to discuss the SSD and the risk of weight gain in Discussion Section. Please see Line 262-266 in the revised manuscript.

Reviewer 2 Report
very good study
- The study convey the gap of knowledge on obesity and effort rewards and diabetes in adults related to work related stress. They find that short sleep duration and low effort rewards are main independent indicators for development of obesity and diabetes. The results are new and they can have pioneering impact on how to help prevention on obesity and diabetes.
- The strengths of the study is that better sleep and better effort rewards at work is possible to implement in the workplaces to prevent the global epidemy of obesity and diabetes. The weakness is that this is confirmed in in one study among one type of job so following studies are needed to confirm the results again in different jobs.
- Even the authors have done a great work to explain the theoretical background I still think the discussion should be shortened by 25 or 30%. It is a little too hard to come through the whole thing and by taking this 30% out, the conclusion should shortly say what has this study contributed with and which other follow-up studies are recommended to try out this findings as hypothesis to confirm the results in different jobs.
Author Response
Point 1: The study convey the gap of knowledge on obesity and effort rewards and diabetes in adults related to work related stress. They find that short sleep duration and low effort rewards are main independent indicators for development of obesity and diabetes. The results are new and they can have pioneering impact on how to help prevention on obesity and diabetes.
Response 1: Thank you for your suggestion. We highlighted the novel point to prevent the incidence of obesity and T2DM. Please see Line 303-304 in the revised manuscript.
Point 2: The strengths of the study is that better sleep and better effort rewards at work is possible to implement in the workplaces to prevent the global epidemy of obesity and diabetes. The weakness is that this is confirmed in in one study among one type of job so following studies are needed to confirm the results again in different jobs.
Response 2: Thank for your suggestion. We totally agree with your comment. Because our study population were civil servants, our findings may not apply to other occupations with heavy workload and repetitive jobs. It is possible to increase the risk of adverse effects. Consequently, our findings may underestimate the results that SSD and high ERI led to high prevalence of obesity and T2DM. Please see Line 287-289 in the revised manuscript
Point 3: Even the authors have done a great work to explain the theoretical background I still think the discussion should be shortened by 25 or 30%. It is a little too hard to come through the whole thing and by taking this 30% out, the conclusion should shortly say what has this study contributed with and which other follow-up studies are recommended to try out this findings as hypothesis to confirm the results in different jobs.
Response 3: Thank you for your suggestion. The discussion section has been amended to a shortened version and the conclusion also concisely emphasizes our findings. Paragraph 1 in the discussion section has been removed and thus become more concise in the discussion section. The hypothesis is needed to be verified by prospective studies. Please see Line 309 in the revised manuscript.

Reviewer 3 Report
Review: Sleep Duration and Effort-Reward Imbalance (ERI) 2 Associated with Obesity and Type II Diabetes 3 Mellitus (T2DM) among Taiwanese Middle-aged 4 Public Servants
The article addresses a topic of interest in public health and occupational health. Thus, the article analyzes the health problem addressed in a perspective that is still little explored, and that is promising. Obesity and Type 2 Diabetes Mellitus represent global health problems in an increasing trend. Analyzing the contribution of occupational stressors and sleep duration and quality can play an essential role in interventions to reduce this occurrence as described in the text
The objective of the study is established: “The objective of this study is to assess the interaction effects of sleep duration and effort-reward imbalance (ERI) and its correlation with the risk of obesity and T2DM among Taiwanese public servants aged 40”. As the study has a cross-sectional design cross-sectional study, the use of the term “risk” seems to be inappropriate. I suggest revising the term (replacing it by frequency or merely deleting the term from the objective. Even without it, there will be a logical sense in the writing of the objectives).
Considering the proposed objective, the purpose of T2DM analysis based on obesity and overweight strata (presented in results) is not clear. What is this analysis for? What is its purpose?
Materials at Methods.
The information about the study population needs to be more precise and informative. The total number of workers is presented (“A total of 11,875 participants from 647 registered governmental institutions”), without specifying whether this N refers to the total number of workers in the institutions selected for the study or refers to the N in the defined age group ( 40 to 60 years)? It is necessary to present the data directly and clearly. It is important to note that the data in Table 1, the total N is 10,795; therefore, there is a difference of 1,080 workers. Clarify this data.
Results
It is necessary to reformulate the phrase "Levels of sleep duration were significantly correlated with the OR of T2DM". OR is a measure of association and expresses the relationship between two events, in this case, 'levels of sleep duration and T2DM'. Therefore, levels of sleep duration are not related to OR, but to T2DM - OR is a measure that expresses the magnitude of this correlation. Review the wording throughout this paragraph (e.g., "People with SSD for both workday and weekend / vacation (level 4) had 1.48 times OR of T2DM").
Table 2 shows in the title "Two groups of sleep duration for workday and weekend/vacation and correlation with obesity, overweight, BMI and effort-reward imbalance (ERI)." There is a reference to correlation in the title, but the data presented in the table do not refer to "correlation": they are percentage distribution of the outcomes under analysis according to the two groups of sleep duration. Remove the term correlation from the title and presentation of the text. There is also no test information of statistical significance or p values ​​in table 2, but in the text, it is mentioned, "Levels of BMI were significantly correlated with sleep duration" (p.3, line 126). As the term "significantly" is traditionally used to express reference of statistical significance, it is more appropriate to use other terminology or to present data of statistical significance - if you have performed such tests.
It is interesting to highlight the very significant differences in the ERI distribution according to the level of sleep duration for workday and weekend / vacation.
Table 3 shows the letters a and c, but there is no note to which they refer. Complete the information. Apparently, one analysis is crude, and the other adjusted, but you need to specify what it refers to. Asterisks (**) are also missing information about which they refer. It is also necessary to inform, in the table, the levels from 1 to 4 - although it is in the text, it is advisable to inform in the table Tables must be self-explanatory.
As I asked above, what is the reason for doing this analysis “Table 4 shows correlations between sleep duration for workday and weekend / vacation and 140 T2DM in non-obese and obese groups using multiple logistic regression”? What question does this analysis wish to answer? Apparently, it is a question different from the one contained in the mentioned objective (which, if I understand correctly, is to evaluate the effects of sleep duration and effort-reward imbalance (ERI) interaction in the frequency of obesity and T2DM). If it is a different question, this other question and purpose must be indicated in the objectives.
Is it objective to evaluate the relationship between the two outcomes investigated, obesity and T2DM? Why consider T2DM in obesity / overweight strata?
The same question arises for the data in Tables 5 and 6.
Again, I suggest caution with the use of the term “risk” (p.4, line 149: “Table 5 indicates that risks of T2DM”).
Again, I call attention to the restricted use of terms - on p. 4, line 155-156, ”Clearly, there are interaction effects of BMI and sleep duration on the OR of T2DM“. In epidemiology, there are specific procedures for evaluating interaction (be it multiplicative or additive). Results that show the frequency of an event increasing or decreasing due to the exposure of a variable (exposure), even with a tendency of dose-response effect, stricto sensu, cannot be called “interaction” (the interaction is observed when the presence two exposures produce more cases than expected due to excess or multiplication of their separate effects). What was observed in the case was an association between the investigated events, with a dose-response effect.
It is suggested that the use of dose-response trend assessment tests (Mantel-Haenszel chi-square for trend analysis, for example) confirms this hypothesis.
Data from Table 6 reinforce the doubts already mentioned in Tables 4 and 5. It is not clear precisely the question that one wishes to answer when evaluating T2MD in obesity and overweight strata. It is necessary, initially, to specify this in the text to inform and justify this. In this way, the analysis of these tables will be more clearly understood.
Table 7 shows data that is related to the proposed objective. The question asked in the objectives can be answered by the data presented in this table regarding obesity and overweight, but the T2DM data remains stratified by groups of obese and non-obese individuals. Why was the direct relationship between ERI and sleep duration not evaluated with T2DM?
In the end, the article looks more like a combined exposure association analysis than an interaction effect study (as mentioned in the text). The only method most traditionally used in epidemiology and used to assess interaction was the synergy index.
The analysis of compound exposure is fully justified and provides beneficial information. Therefore, it seems that treating the study in these terms is more appropriate than treating it as an interaction study - since more specific interaction assessment tools were not employed.
Discussion
Page 7, lines from 196 to 211 should be relocated to other sections. There is no discussion of data in the text; it is information about the study population and the way of selecting the sample.
Therefore, it must be redirected to the Material and Methods section or the beginning of the Results item.
In the lines from 212 to 245, references are cited that bring general content - which points to supporting data for the relevance of the study carried out, the controversies still present, among other themes. Thus, they are more appropriate for the introduction section than the discussion section. It is important to remember that this section should discuss the results produced in the Results section. The data brought in these lines can be cited as long as they make a direct dialogue with the results obtained. Therefore, the revision of this excerpt is necessary.
Page, lines 246-247, "Our findings assessed the relationship between sleep duration and job strain on the risk of T2DM 246 after adjusting for obesity.". Here again, it is worth asking why to adjust for obesity? Caution with the use of "job strain" - analysis was made with the ERI and not with the JCQ (job strain, in general, is used for the Demand-Control model). I suggest keeping the term job stress used.
Page 10, lines 332-334 - Review this. (Acknowledgments: In this section you can acknowledge any support given which is not covered by the author 332 contribution or funding sections. This may include administrative and technical support, or donations in kind 333 (e.g., materials used for experime).
Conclusion
The article brings significant contributions. The authors studied a national sample with a significant sample size and explored still controversial subjects in the literature. Therefore, I consider that the article, after major revision, can be published.
Points for review: 1. explain the reason for assessing the relationship between T2DM and obesity (it is not explained in the objective of the study, as described in the current format); 2. Maintain caution in the use of terms that are not adequate or with inaccuracies (such as “risk,” “OR of T2DM”, correlation); 3. Reassess whether the study evaluates interaction or double exposure / combined exposure.
Author Response
Point 1: The objective of the study is established: “The objective of this study is to assess the interaction effects of sleep duration and effort-reward imbalance (ERI) and its correlation with the risk of obesity and T2DM among Taiwanese public servants aged 40”. As the study has a cross-sectional design cross-sectional study, the use of the term “risk” seems to be inappropriate. I suggest revising the term (replacing it by frequency or merely deleting the term from the objective. Even without it, there will be a logical sense in the writing of the objectives).
Considering the proposed objective, the purpose of T2DM analysis based on obesity and overweight strata (presented in results) is not clear. What is this analysis for? What is its purpose?
Response 1: Thank you for your comment. The objective of this study is to assess the moderating effects of sleep duration and effort-reward imbalance (ERI) on the prevalence of obesity and T2DM among Taiwanese public servants aged 40 to 60. People with short sleep duration and high ERI were population susceptible to obesity and T2DM development. The mechanism of sleep duration or job stress on T2DM could alter appetite regulation, increase body weight, decrease glucose tolerance, increase blood pressure, and increase insulin resistance and sympathetic tone. Please see Line 63-65 and 262-266 in the revised manuscript.
Point 2: Materials at Methods.
The information about the study population needs to be more precise and informative. The total number of workers is presented (“A total of 11,875 participants from 647 registered governmental institutions”), without specifying whether this N refers to the total number of workers in the institutions selected for the study or refers to the N in the defined age group (40 to 60 years)? It is necessary to present the data directly and clearly. It is important to note that the data in Table 1, the total N is 10,795; therefore, there is a difference of 1,080 workers. Clarify this data.
Response 2: Thank you for your comment. You are right. A total of 11,875 participants responded to web-based questionnaire, but 1080 subjects did not comprehensively fill out the part of the question about sleep duration due to insufficient time and oblivion. The difference of subjects may not relate to the objective of this study.
Point 3: Results
It is necessary to reformulate the phrase "Levels of sleep duration were significantly correlated with the OR of T2DM". OR is a measure of association and expresses the relationship between two events, in this case, 'levels of sleep duration and T2DM'. Therefore, levels of sleep duration are not related to OR, but to T2DM - OR is a measure that expresses the magnitude of this correlation. Review the wording throughout this paragraph (e.g., "People with SSD for both workday and weekend / vacation (level 4) had 1.48 times OR of T2DM").
Response 3: Thank you for your comment. The phrase has been amended to “levels of sleep duration were significant associated with percentage of T2DM”. In addition, the phrase has also been changed to "People with SSD for both workday and weekend/vacation (level 4) had 1.48 times higher risks of T2DM" (OR=1.48)). Please see Line 140-141 in the revised manuscript.
Point 4: Table 2 shows in the title "Two groups of sleep duration for workday and weekend/vacation and correlation with obesity, overweight, BMI and effort-reward imbalance (ERI)." There is a reference to correlation in the title, but the data presented in the table do not refer to "correlation": they are percentage distribution of the outcomes under analysis according to the two groups of sleep duration. Remove the term correlation from the title and presentation of the text. There is also no test information of statistical significance or p values in table 2, but in the text, it is mentioned, "Levels of BMI were significantly correlated with sleep duration" (p.3, line 126). As the term "significantly" is traditionally used to express reference of statistical significance, it is more appropriate to use other terminology or to present data of statistical significance - if you have performed such tests.
It is interesting to highlight the very significant differences in the ERI distribution according to the level of sleep duration for workday and weekend / vacation.
Response 4: Thank you for your comment. We have modified “correlation” to “association” and "significantly" to "statistical significance".
Point 5: Table 3 shows the letters a and c, but there is no note to which they refer. Complete the information. Apparently, one analysis is crude, and the other adjusted, but you need to specify what it refers to. Asterisks (**) are also missing information about which they refer. It is also necessary to inform, in the table, the levels from 1 to 4 - although it is in the text, it is advisable to inform in the table Tables must be self-explanatory.
Response 5: Response: Thank you for your suggestions. We have changed “c” and “a” to “Crude” and “Adjusted”. Asterisks (**) has been explained in All Tables in the revised manuscript. We have added information to define the levels of sleep duration in Table 3. Please see Line 166-169 in the revised manuscript.
Point 6: As I asked above, what is the reason for doing this analysis “Table 4 shows correlations between sleep duration for workday and weekend / vacation and 140 T2DM in non-obese and obese groups using multiple logistic regression”? What question does this analysis wish to answer? Apparently, it is a question different from the one contained in the mentioned objective (which, if I understand correctly, is to evaluate the effects of sleep duration and effort-reward imbalance (ERI) interaction in the frequency of obesity and T2DM). If it is a different question, this other question and purpose must be indicated in the objectives.
Is it objective to evaluate the relationship between the two outcomes investigated, obesity and T2DM? Why consider T2DM in obesity / overweight strata?
The same question arises for the data in Tables 5 and 6.
Again, I suggest caution with the use of the term “risk” (p.4, line 149: “Table 5 indicates that risks of T2DM”).
Response 6: Thank you for your comments. Previous studies have reported that obesity is significantly associated with T2DM. Since our study stratified BMI into obese and non-obese groups, a consistent finding was confirmed to indicate that sleep hours in workday are associated with T2DM by using multiple logistic regression adjusted for covariates in Table 4. Similarly, Table 7 showed that interaction effect of ERI and sleep duration during the workday on overweight/obesity and T2DM using multiple logistic regression after adjusting for covariates. Our study has assessed the association between obesity and T2DM when stratified by sleep duration and ERI in Tables 5 and 6. We found different effects on obesity and T2DM in the two groups with sleep duration and ERI, indicating that sleep duration and ERI might play a role of modification effect. Because “risk” of T2DM in the cross-sectional study was used, we have amended the term “risk” to be replaced by prevalence of T2DM in the revised manuscript.
Point 7: Again, I call attention to the restricted use of terms - on p. 4, line 155-156, ”Clearly, there are interaction effects of BMI and sleep duration on the OR of T2DM“. In epidemiology, there are specific procedures for evaluating interaction (be it multiplicative or additive). Results that show the frequency of an event increasing or decreasing due to the exposure of a variable (exposure), even with a tendency of dose-response effect, stricto sensu, cannot be called “interaction” (the interaction is observed when the presence two exposures produce more cases than expected due to excess or multiplication of their separate effects). What was observed in the case was an association between the investigated events, with a dose-response effect.
It is suggested that the use of dose-response trend assessment tests (Mantel-Haenszel chi-square for trend analysis, for example) confirms this hypothesis.
Response 7: Thank you for your comments. Interaction effects were assessed by Synergy index, which represented additive interaction of ERI and sleep duration during the workday on overweight/obesity and T2DM. No dose-response trend was tested in the study because the binary variable of sleep duration and ERI was measured. Certainly, Mantel-Haenszel chi-square for trend analysis is not necessary to be tested in the study.
Point 8: Data from Table 6 reinforce the doubts already mentioned in Tables 4 and 5. It is not clear precisely the question that one wishes to answer when evaluating T2MD in obesity and overweight strata. It is necessary, initially, to specify this in the text to inform and justify this. In this way, the analysis of these tables will be more clearly understood.
Table 7 shows data that is related to the proposed objective. The question asked in the objectives can be answered by the data presented in this table regarding obesity and overweight, but the T2DM data remains stratified by groups of obese and non-obese individuals. Why was the direct relationship between ERI and sleep duration not evaluated with T2DM?
Response 8: Thank you for your comments. Based on the previous studies, it is indicated that obesity leads to T2DM development. Our study was stratified into the two groups of sleep duration and ERI to assess the difference on association between obesity and T2DM in Tables 5 and 6. We found different effects on obesity and T2DM with sleep duration and ERI in the two groups, showing that sleep duration and ERI might play a role of modification effect. Similarly, Table 7 displayed the interaction effect of ERI and sleep duration during the workday on overweight/obesity and T2DM using multiple logistic regression after adjusting for covariates.
Point 9: In the end, the article looks more like a combined exposure association analysis than an interaction effect study (as mentioned in the text). The only method most traditionally used in epidemiology and used to assess interaction was the synergy index.
The analysis of compound exposure is fully justified and provides beneficial information. Therefore, it seems that treating the study in these terms is more appropriate than treating it as an interaction study - since more specific interaction assessment tools were not employed.
Response 9: Thank you for your comments. We totally agree with your comments about using the interaction assessment tools in the study. Further researches are needed to elaborate on the interaction effect of ERI or sleep duration and T2DM by mediating the effect of obesity. The direct and indirect effects of ERI or sleep duration on prevalence of obesity and T2DM should be assessed. Please see Line 297-299 in the revised manuscript.
Point 10: Discussion
Page 7, lines from 196 to 211 should be relocated to other sections. There is no discussion of data in the text; it is information about the study population and the way of selecting the sample.
Therefore, it must be redirected to the Material and Methods section or the beginning of the Results item.
Response 10: Thank you for your suggestions. Lines from 196 to 211 have been relocated to Methods.
Point 11: In the lines from 212 to 245, references are cited that bring general content - which points to supporting data for the relevance of the study carried out, the controversies still present, among other themes. Thus, they are more appropriate for the introduction section than the discussion section. It is important to remember that this section should discuss the results produced in the Results section. The data brought in these lines can be cited as long as they make a direct dialogue with the results obtained. Therefore, the revision of this excerpt is necessary.
Response 11: Thank you for your suggestions. Lines from 212 to 245 was refined and became more concise in illustration. The Paragraph indicated the findings of sleep duration and job stress associated with T2DM compared to the previous studies.
Point 12: Page, lines 246-247, "Our findings assessed the relationship between sleep duration and job strain on the risk of T2DM 246 after adjusting for obesity.". Here again, it is worth asking why to adjust for obesity? Caution with the use of "job strain" - analysis was made with the ERI and not with the JCQ (job strain, in general, is used for the Demand-Control model). I suggest keeping the term job stress used.
Response 12: Thank you for your suggestions. You are right. We use ERI to measure job stress in the study. In the revised manuscript the term has been changed from “job strain” to “job stress”. Please see the revised manuscript.
Point 13: Page 10, lines 332-334 - Review this. (Acknowledgments: In this section you can acknowledge any support given which is not covered by the author 332 contribution or funding sections. This may include administrative and technical support, or donations in kind 333 (e.g., materials used for experime).
Response 13: Thank you for your suggestions. Revised manuscript has been amended to acknowledge all administrative and technical supports from central and local governments. Please see Line 321-323 in the revised manuscript.
Point 14: Conclusion
The article brings significant contributions. The authors studied a national sample with a significant sample size and explored still controversial subjects in the literature. Therefore, I consider that the article, after major revision, can be published.
Response 14: Thank you for your comments. We made efforts to amend some points of the controversial subjects to make them fluent to read. Please see the revised manuscript.
Point 15: Points for review: 1. explain the reason for assessing the relationship between T2DM and obesity (it is not explained in the objective of the study, as described in the current format); 2. Maintain caution in the use of terms that are not adequate or with inaccuracies (such as “risk,” “OR of T2DM”, correlation); 3. Reassess whether the study evaluates interaction or double exposure / combined exposure.
Response 15: Thank you for your suggestions.
- We have added information to illustrate the objective of assessing the interaction effects of sleep duration and effort-reward imbalance (ERI) on the risk of obesity and T2DM.
- “Risk” term or “OR of T2DM” have been changed to “prevalence of T2DM”.
- Because the study has been classified into the two groups of sleep duration and ERI, additive interaction of sleep duration and ERI on prevalence of obesity and T2DM was assessed by synergy index. Further research is crucial to elaboration of the mediation analysis for sleep duration or ERI on prevalence and T2DM by obesity.

Round 2
Reviewer 1 Report
The revision has improved the MS. I have no further concern.
Author Response
We appreciated your comments.
Reviewer 3 Report
Review 2
The new version includes a revision of some points of the requested previous notes. The current version has made significant improvements in the writing of the article. However, relevant problems remain:
1. The Introduction discusses possible relationships between work characteristics (ERI) and sleep duration on obesity and type 2 diabetes mellitus (T2DM). However, it still does not discuss obesity and T2DM. This aspect is not in the Introduction or Methods. If "Previous studies have reported that obesity is significantly associated with T2DM" (as in the authors' response), it is a pertinent discussion to include in the Introduction, for example, or in Methods. Nevertheless, there was no change in the text regarding this vital aspect.
There is still a gap about the reasons that guided the analysis of the association between sleep duration and ERI and T2DM, considering obesity groups. The underlying hypothesis (which can be assumed) is that these outcomes are also related to each other - so the combination of these two outcomes increases health risks. If this is a hypothesis, it is necessary to make it explicit (I think this is logical and can structure possibilities for analyzing the data, but a clear definition of this is necessary). Alternatively, if it was another reason that produced this analytical path - what motivated the choice - to be presented.
The authors analyzed T2Dm considering two groups of obesity (tables 4 and 7). The direct relationship between work characteristics (ERI) or sleep duration and type 2 diabetes mellitus (T2DM) has not been performed (as done for obesity). These differentiated procedures for the two outcomes analyzed need to be clarified. What motivated this analytical decision is still unanswered.
In summary, my note was that, given the relevance that this analysis has in the article, this needs to be well defined in the construction of the article (for example, the authors should mention the importance and justification of this analysis).
2. Regarding the study population, the original information is unchanged in the new version. In Methods, the inclusion of 11,875 people is reported, while Results show an analysis of only 10,795 people. Therefore, the final population studied was 10,795 (Results data) - this is the basis for recalculating the response rate (which will not be 35.8%). If 1,080 did not answer the study questions and are not included in the analysis, it does not constitute the study population.
An analysis of the comparison between respondents and non-respondents was not requested. Thus, the authors' reply that "The difference of subjects may not report to the objective of this study" does not answer my question about it. In the Discussion item (limits of the study) should include this aspect.
Therefore, it is necessary to review: the total number of workers analyzed/included in the study; estimate the response rate again; discuss the implications of a survey with a response rate of 32-36% of the target population in the "Discussion" section.
3. The presentation of the results of the association measures (OR) remained with problems. Ex: "Table 3 indicates associations between sleep duration during the workday and weekend/vacation and the ORs of T2DM using multiple logistic regression [...]".
Therefore, the request to review this has not been made [I forward my previous comments below). OR is a measure of association; therefore, it does not make sense to say the association of [...] with OR (association of a variable with the association measure?].
Example: There was a significant association between sleep duration for workday and ORs of [this is not correct) T2DM (OR = 1.48 for the obese group and OR = 1.39 for the non-obese group, respectively), but not for weekend / vacation - should be revised to “There was a significant association between sleep duration for workday and T2DM (OR = 1.48 for the obese group and OR = 1.39 for the non-obese group, respectively) [...].
Comment in the previous review: Point 15: Points for review: 1. explain the reason for assessing the relationship between T2DM and obesity (it is not explained in the objective of the study, as described in the current format); 2. Maintain caution in the use of terms that are not adequate or with inaccuracies (such as “risk,” “OR of T2DM”, correlation); 3. Reassess whether the study evaluates interaction or double exposure / combined exposure.
4. If “No dose-response trend was tested in the study because the binary variable of sleep duration and ERI was measured,” on the basis that it is mentioned that “Levels of BMI were 160 dose-dependently associated with ORs of T2DM development in both groups of SSD and LSD ”(p.4, line 161).
Final recommendation: Although the current version has improved the clarity and conceptual correction of the article, there are still significant inaccuracies and gaps - already pointed out in the previous review - that have not been adequately addressed. Therefore, I do not recommend the approval of the article in its current format.
Author Response
The new version includes a revision of some points of the requested previous notes. The current version has made significant improvements in the writing of the article. However, relevant problems remain:
Point 1: The Introduction discusses possible relationships between work characteristics (ERI) and sleep duration on obesity and type 2 diabetes mellitus (T2DM). However, it still does not discuss obesity and T2DM. This aspect is not in the Introduction or Methods. If "Previous studies have reported that obesity is significantly associated with T2DM" (as in the authors' response), it is a pertinent discussion to include in the Introduction, for example, or in Methods. Nevertheless, there was no change in the text regarding this vital aspect.
Response: Thank you for your comments. Our manuscript highlighted that ERI and sleep duration on both effects of obesity and T2DM. Therefore, interaction effect of ERI and sleep duration on effects of overweight/obesity and T2DM using multiple logistic regression after adjusting for covariates was shown in Table 7. Previous studies have indicated that overweight/obesity significantly associated with T2DM. However, we should elaborate the direct or indirect effect between ERI and SSD on T2DM by mediating effect of obesity. In revised manuscript has added to limitation in discussion. Further research is needed to use the mediation and moderation analysis for effect of ERI and SSD on obesity and T2DM.
There is still a gap about the reasons that guided the analysis of the association between sleep duration and ERI and T2DM, considering obesity groups. The underlying hypothesis (which can be assumed) is that these outcomes are also related to each other - so the combination of these two outcomes increases health risks. If this is a hypothesis, it is necessary to make it explicit (I think this is logical and can structure possibilities for analyzing the data, but a clear definition of this is necessary). Alternatively, if it was another reason that produced this analytical path - what motivated the choice - to be presented.
Response: Thank you for your suggestive comments. You are right. Because obesity significantly associated with T2DM, our study cannot determine the ERI and SSD lead to obesity and then result in T2DM. It is needed to use structure equation model (SEM) for path analysis of between ERI and SSD on T2DM by mediating effect of obesity. The model can explicit that indirect and direct effect of ERI and SSD on obesity and then lead to T2DM. Therefore, the objective of this study only focused on that interaction effect of ERI and SSD on obesity and T2DM, but not shown the mediation effect of obesity for ERI and SSD on T2DM.
The authors analyzed T2Dm considering two groups of obesity (tables 4 and 7). The direct relationship between work characteristics (ERI) or sleep duration and type 2 diabetes mellitus (T2DM) has not been performed (as done for obesity). These differentiated procedures for the two outcomes analyzed need to be clarified. What motivated this analytical decision is still unanswered.
Response: Table 4 only focused on the association between SSD and T2DM in obese and non-obese groups using multivariate analysis adjusted for covariates. The results indicated that SSD significantly associated with T2DM in the two groups of obesity, but the findings did not involve ERI variable in the model. However, Table 7 found that interaction effect of ERI and SSD during the workday on overweight/obesity and T2DM using multivariate analysis adjusted for covariates. There were different purposes to describe the association between SSD and ERI with obesity and T2DM for two Tables.
In summary, my note was that, given the relevance that this analysis has in the article, this needs to be well defined in the construction of the article (for example, the authors should mention the importance and justification of this analysis).
Response: Thank you for your comments. The significance of this study is to find the interaction effect of SSD and ERI on obesity and T2DM. In addition, our findings indicated the association between obesity and T2DM by moderating effect of ERI or sleep duration. However, the findings cannot confirm the casual pathway between SSD and ERI with direct effect or indirect effect of obesity and T2DM.
Point 2: Regarding the study population, the original information is unchanged in the new version. In Methods, the inclusion of 11,875 people is reported, while Results show an analysis of only 10,795 people. Therefore, the final population studied was 10,795 (Results data) - this is the basis for recalculating the response rate (which will not be 35.8%). If 1,080 did not answer the study questions and are not included in the analysis, it does not constitute the study population.
Response: Thank you for your suggestion. We have changed the study population by 10,795 and response rate in revised manuscript. Please see line 71-72 in revised manuscript.
An analysis of the comparison between respondents and non-respondents was not requested. Thus, the authors' reply that "The difference of subjects may not report to the objective of this study" does not answer my question about it. In the Discussion item (limits of the study) should include this aspect.
Therefore, it is necessary to review: the total number of workers analyzed/included in the study; estimate the response rate again; discuss the implications of a survey with a response rate of 32-36% of the target population in the "Discussion" section.
Response: Thank you for your suggestion. We have added to illustrate no difference of demographic and work characteristics for respondents and non-respondents in Discussion section. Please see line 287-290 in revised manuscript.
Point 3: The presentation of the results of the association measures (OR) remained with problems. Ex: "Table 3 indicates associations between sleep duration during the workday and weekend/vacation and the ORs of T2DM using multiple logistic regression [...]".
Therefore, the request to review this has not been made [I forward my previous comments below). OR is a measure of association; therefore, it does not make sense to say the association of [...] with OR (association of a variable with the association measure?].
Example: There was a significant association between sleep duration for workday and ORs of [this is not correct) T2DM (OR = 1.48 for the obese group and OR = 1.39 for the non-obese group, respectively), but not for weekend / vacation - should be revised to “There was a significant association between sleep duration for workday and T2DM (OR = 1.48 for the obese group and OR = 1.39 for the non-obese group, respectively) [...].
Response: Thank you for your suggestions. We have amended the sentence in the revised manuscript. Please see line 150-152 in revised manuscript.
Comment in the previous review: Point 15: Points for review: 1. explain the reason for assessing the relationship between T2DM and obesity (it is not explained in the objective of the study, as described in the current format); 2. Maintain caution in the use of terms that are not adequate or with inaccuracies (such as “risk,” “OR of T2DM”, correlation); 3. Reassess whether the study evaluates interaction or double exposure / combined exposure.
Response: Thank you for your comments. Based on your previous review we have added or amended in revised manuscript.
- We have added to illustrate the objective for assessing the interaction effects of sleep duration and effort-reward imbalance (ERI) on obesity and T2DM. In addition, the study will assess the association between obesity and T2DM by moderating effect of ERI or sleep duration. Please see line 65-66 in revised manuscript.
- “Risk” term or “OR of T2DM “have changed to” prevalence of T2DM”.
- Because the study has classified two groups of sleep duration and ERI, additive interaction of sleep duration and ERI on prevalence of obesity and T2DM was assessed by synergy index. Nevertheless, the path analysis should be used to elaborate the causal pathway of ERI and SSD on obesity and T2DM or assess the mediation analysis for sleep duration or ERI on T2DM by obesity using structure equation model (SEM). Please see line 302-304 in revised manuscript.
Point 4: If “No dose-response trend was tested in the study because the binary variable of sleep duration and ERI was measured,” on the basis that it is mentioned that “Levels of BMI were 160 dose-dependently associated with ORs of T2DM development in both groups of SSD and LSD ”(p.4, line 161).
Response: We have amended the sentence in revised manuscript. Please see line 160-161 in revised manuscript.
Final recommendation: Although the current version has improved the clarity and conceptual correction of the article, there are still significant inaccuracies and gaps - already pointed out in the previous review - that have not been adequately addressed. Therefore, I do not recommend the approval of the article in its current format.
Response: Thank you for your comments and suggestions. Our manuscript has been improved based on your suggestions. Please see in revised manuscript.

This manuscript is a resubmission of an earlier submission. The following is a list of the peer review reports and author responses from that submission.